# The Defensive Role of Endogenous H_2_S in *Brassica rapa* against Mercury-Selenium Combined Stress

**DOI:** 10.3390/ijms23052854

**Published:** 2022-03-05

**Authors:** Lifei Yang, Huimin Yang, Zhiwei Bian, Haiyan Lu, Li Zhang, Jian Chen

**Affiliations:** 1Department of Horticulture, College of Horticulture, Nanjing Agricultural University, Nanjing 210095, China; lfy@njau.edu.cn (L.Y.); 2019104074@njau.edu.cn (H.Y.); 201410408@njau.edu.cn (Z.B.); 2Hexian New Countryside Development Research Institute, Nanjing Agricultural University, Hexian 238200, China; 3Laboratory for Food Quality and Safety-State Key Laboratory Cultivation Base of Ministry of Science and Technology, Institute of Food Safety and Nutrition, Jiangsu Academy of Agricultural Sciences, Nanjing 210014, China; luhaiyan8282@163.com; 4Department of Tobacco, College of Plant Protection, Shandong Agricultural University, Taian 271018, China; lilizhang324@163.com

**Keywords:** *Brassica rapa*, hydrogen sulfide, mercury, reactive oxygen species, selenium

## Abstract

Plants are always exposed to the environment, polluted by multiple trace elements. Hydrogen sulfide (H_2_S), an endogenous gaseous transmitter in plant cells, can help plant combat single elements with excess concentration. Until now, little has been known about the regulatory role of H_2_S in response to combined stress of multiple elements. Here we found that combined exposure of mercury (Hg) and selenium (Se) triggered endogenous H_2_S signal in the roots of *Brasscia rapa*. However, neither Hg nor Se alone worked on it. In roots upon Hg + Se exposure, the defensive role of endogenous H_2_S was associated to the decrease in reactive oxygen species (ROS) level, followed by alleviating cell death and recovering root growth. Such findings extend our knowledge of plant H_2_S in response to multiple stress conditions.

## 1. Introduction

Heavy metal pollution poses a threat to biological systems, which is a serious problem worldwide. Heavy metal exposure impacts plant growth and development [1]. Hg (mercury) is one of the toxic metals even at low levels. Hg content increases continuously in the environment because of natural release and anthropogenic activities. The average content of soil Hg is 1.1 mg/kg worldwide. The polluted soils contain Hg at various levels even more than 10 mg/kg or higher [2]. Hg^2+^ is the most toxic form of Hg. Hg^2+^ can be readily taken up by plants to induce phytotoxicity. In hydroponic studies, plants exposed to levels of Hg^2+^ (even at 1–5 µM) begin to exhibit toxic symptoms [3,4]. Hg-induced phytotoxicity is associated with the accumulation of ROS (reactive oxygen species), followed by oxidative stress, physiological disturbance, and cell death [3,5]. ROS accumulation is considered as bio-indicator of Hg-induced phytotoxicity [4].

Se (selenium) is a beneficial micronutrient for plant growth. Se at proper concentrations, e.g., 0.5–6.0 mg/L SeO_3_^2−^ (about 3.9–47 µM Se) can rescue plant from the toxicity of heavy metals including Hg [6]. However, Se application with excess concentration shows synergistic toxic effect with Hg [7]. Excessive Se is an emerging environmental problem [8]. Se and Hg frequently coexist in the environment [9]. Our previous study also found synergistic toxic effect of Se and Hg on *Brassica rapa*, showing enhanced ROS accumulation and aggravated growth inhibition as compared to Se or Hg alone [10]. However, we still rarely know the plant signaling regulation in response to Hg + Se combined stress.

H_2_S (hydrogen sulfide) is an endogenous signaling molecule in plant cells. H_2_S can be generated during sulfate assimilation. DES (_L_-cysteine desulfhydrase, EC4.4.1.1) is considered to be the main enzyme to produce H_2_S, regulating various plant physiological processes [11]. H_2_S plays important roles in the modulation of plant development and stress responses [12]. Few studies focus on the function of H_2_S in plants upon multiple stress conditions at the same time. H_2_S regulates plant tolerance against heavy metal stress, such as Cd (cadmium) [13], Al (aluminum) [14], Cr (chromium) [15], Pb (lead) [16], and Hg [17], etc. These reports suggest that H_2_S protects plants from metal toxicity by alleviating oxidative injury. We previously found that H_2_S was essential for plant combating excessive Se stress [18]. However, whether and how endogenous H_2_S regulates plant response upon Se + Hg combination remains elusive.

In this study, we identified the intensified H_2_S signaling in the root of *B. rapa* under Hg + Se stress conditions. The results not only help reveal plant signaling regulation of synergistic toxicity of Hg and Se, but also extend our knowledge of H_2_S in plant cells in response to multiple stress conditions. 

## 2. Results and Discussion

The roots of *B. rapa* were exposed to HgCl_2_ (mercury chloride) (1 µM), Na_2_SeO_3_ (sodium selenite) (4 µM), or their combination for 3 days. Treatment with Hg alone slightly inhibited root growth (Figure 1A). This confirms the phytotoxic effect of Hg even at low levels [19]. Se induces phytotoxicity only at high concentrations [20]. Here we found that treatment with Se alone failed to affect root growth (Figure 1A), suggesting that Se at 4 µM was safe for the growth of *B. rapa* seedlings. However, simultaneous treatment with Hg and Se resulted in remarkable decrease in root length by 48.2% as compared to the control (Figure 1A). Since Hg + Se, but not Hg or Se alone, impacted root growth, we further performed time-course experiment to monitor root length upon Hg + Se exposure. Root growth was significantly inhibited after treatment with Hg + Se for 6 h. Then root growth speed declined with prolonged Hg + Se treatment (Figure 1B). Either Hg or Se was relative safe for the growth of *B. rapa* (Hg showed slight toxicity), but Hg + Se induced severe phytotoxicity. Treatment with Hg and Se together may overwhelm toxic threshold of single element, which further impacting plant growth. Therefore, Se showed synergistic but not antagonistic effect on Hg stress. 

The elongation of root tip determines root growth in response to environmental stress [21]. H_2_S plays defensive roles in plants against environmental stress [12]. To study the response of H_2_S in root tip, we used specific probe WSP-1 (3′-methoxy-3-oxo-3H-spiro[isobenzofuran-1, 9′-xanthen]-6′-yl 2-(pyridin-2-yldisulfanyl)benzoate) to label endogenous H_2_S in situ. Neither Hg nor Se treatment alone affected endogenous H_2_S level in root tip (Figure 2). Hg + Se treatment enhanced endogenous H_2_S remarkably (Figure 2A). WSP-1 fluorescent density increased significantly by 7.71 fold in root tip upon Hg + Se as compared to control (Figure 2B). The changing pattern of endogenous H_2_S was associated with root growth inhibition upon stress conditions. Endogenous H_2_S were stimulated by Hg + Se (inhibiting growth remarkably), but not Hg or Se alone (without remarkable growth inhibition) (Figure 1A and Figure 2B). The increase in endogenous H_2_S level has been found in plant cells in response to growth-inhibited stimuli, such as salinity [22], cadmium (Cd) [23], and chromium (Cr) [24]. Thus, the roots of *B. rapa* may enhance endogenous H_2_S to sense Hg + Se stress. 

Stress-induced endogenous H_2_S production is involved in plant tolerance against abiotic stress [25,26]. Intensified endogenous H_2_S signal help sweet potato seedlings against osmotic stress [27]. To further confirm the defensive role of H_2_S upon Hg + Se stress, we applied pharmacological experiments to alter endogenous H_2_S level in roots. Roots were pretreated with H_2_S donor NaHS (sodium hydrosulfide) for 6 h followed by Hg + Se exposure. Applying NaHS promoted root growth under Hg + Se stress in a dose-dependent manner, with maximal effect occurring at 2 mM NaHS (Figure 3A). Pretreatment with H_2_S scavenger HT (hypotaurine) or DES inhibitor PAG (DL-propargylglicine) aggravated growth retardation upon Hg + Se stress (Figure 3B,C). In addition, HT (4 µM) or PAG (0.5 µM) counteracted the promoting effect of NaHS (2 mM) on root growth upon Hg + Se stress (Figure 3D). In situ fluorescent detection showed that NaHS further enhanced endogenous H_2_S level in root tip treated with Hg + Se, an effect reversed by adding HT or PAG (Figure 3E,F). Hg + Se exposure increased root endogenous H_2_S, but it was not sufficient to help plants combat stress condition. Further enhancing endogenous H_2_S recovered root growth from subsequent Hg + Se stress. An adverse effect was found by decreasing endogenous H_2_S through either scavenging already generated H_2_S or impairing H_2_S biosynthesis. These results suggested a protective role of plant endogenous H_2_S against Hg + Se stress. 

The next step was to study how endogenous H_2_S help plants detoxify Hg + Se. Our previous study found that Hg + Se induced more severe oxidative injury than that of Hg or Se alone in *B. rapa* [10]. We studied the effect of altering endogenous H_2_S on oxidative injury in roots upon Hg + Se stress. Plant development needs to maintain the homeostasis of ROS at proper levels as it is an important signaling molecule. ROS overaccumulation is harmful to plants [28]. Total ROS in root tip were detected in vivo with specific fluorescent probe DCFH-DA (2′,7′-dichlorofluorescein diacetate) [29]. Hg + Se exposure led to remarkable increase in ROS level in root tip as compared to the low level of ROS in control. Pretreatment with NaHS significantly decreased ROS level in root tips under Hg + Se exposure. Further adding PAG or HT resulted in ROS accumulation (Figure 4A,B). These results suggested that enhancing endogenous H_2_S was able to inhibit ROS accumulation induced by Hg + Se stress. 

ROS can cause irreversible cell death in plants in response to stress [30]. Root tip cell death indicated by PI (propidium iodide) showed similar changing pattern to ROS under same treatment conditions (Figure 4C,D), suggesting that enhancing endogenous H_2_S attenuated cell death in root treated with (Hg + Se). Root length was negatively correlated to ROS and cell death, respectively, under the condition of altering endogenous H_2_S level (Figure 4E,F). Therefore, H_2_S-supressed ROS accumulation and cell death may contribute to the recovery of root elongation from Hg + Se stress.

ROS can attack cell membrane lipid, leading to loss of membrane integrity. This is one of the typical consequences of oxidative injury [31]. We applied histochemical staining to evaluate oxidative injury in roots. Lipid peroxidation (indicated by Schiff’s reagent) and loss of membrane integrity (indicated by Evans blue) showed pink and blue, respectively (Figure 5). Hg + Se resulted in extensive staining as compared to light staining in control. Pretreatment with NaHS let to lighter staining in roots treated with Hg + Se, an effect reversed by further adding PAG or HT (Figure 5). These results suggested that endogenous H_2_S was important for plants to combat oxidative induced by Hg + Se stress. Linking the above results provides an important clue that endogenous H_2_S protects plant from Hg + Se stress by suppressing ROS accumulation followed by the alleviation of oxidative injury and cell death. 

The antioxidant role of H_2_S has been associated with its ability of limiting ROS accumulation in response to abiotic stress [32]. H_2_S limits ROS accumulation probably through two ways. First, H_2_S can scavenge already generated ROS by activating both antioxidative enzymes (enzymatic system) and antioxidants (non-enzymatic system) [33]. Second, H_2_S is able to suppress ROS generation. Superoxide radical and hydrogen peroxide are two typical ROS. NADPH oxidase and PAO (polyamine oxidase) generate superoxide radical and hydrogen peroxide, respectively, in plants upon abiotic stress [34,35]. Excessive Se alone induces PAO- and NADPH oxidase-dependent ROS accumulation in plants [36,37]. Hg stress alone can induce NADPH oxidase-dependent ROS burst in plants as well [38]. It has been reported that H_2_S can protect both plant and mammalian cells from oxidative stress by repressing NADPH oxidase-dependent ROS generation [39,40]. Further studies are needed to understand whether H_2_S decreases ROS accumulation in plants against Hg + Se stress through the similar mechanisms mentioned above.

The synergistic toxicity of Hg and Se may directly result from the protein disfunction. The -SH in cystine (Cys) play important roles to maintain functional structure of the proteins. Hg^2+^ has strong affinity to -SH groups in functional proteins [41]. Cys in proteins can be easily replaced by SeCys [8]. Both actions are detrimental to the proteins. The combination of Hg and Se stress may aggravate the disturbance of the structure of functional proteins, further leading to negative physiological responses in plants. Recent studies have revealed that H_2_S improves protein function via persulfidation of Cys [42,43]. Whether H_2_S-mediated protein modification may help overcome the toxicity of Hg + Se needs further studies.

In sum, we found that endogenous H_2_S could be triggered as a defensive signal in response to synergistic toxicity of Hg and Se in *B. rapa*. Neither Hg nor Se alone worked on it. Further studies are needed to reveal the molecular mechanism for H_2_S-mediated detoxification of Hg + Se, our current results propose a novel role of endogenous H_2_S in regulating plant adaption upon multiple stress conditions at the same time. 

## 3. Materials and Methods

### 3.1. Plant Culture, Treatment, and Chemicals

*B. rapa* seeds (Lvling) were obtained from Jiangsu Academy of Agricultural Sciences, China. NaClO (sodium hypochlorite) solution (1%) were used to sterilize the surface of seeds for 10 min. After washing with distilled water, the seeds were soaked with distilled water for 3 h, and allowed for germination in darkness for 1 day. Then the seedlings were transferred into 1/4 strength Hoagland solution. The chamber for seedling growth was set at photoperiod of 12 h, photosynthetic active radiation of 200 μmol/m^2^/s, and temperature at 25 °C. [44]. After growing for another 3 days, we selected 30 identical seedlings with root length at 1.5 cm for each treatment. Different chemicals were added to the nutrient solution to treat seedling roots according to different experimental designs. Basically, the seedlings were treated for 3 days. The seedling only had a primary root without appearing lateral roots. So, we evaluated root growth by only measuring the length of primary root.

Na_2_SeO_3_(sodium selenite) (4 µM) and HgCl_2_ (mercury chloride) (1 µM) were applied for root treatment according to our previous study [10]. NaHS (sodium sulfide) (0.1–16 mM), PAG (DL-propargylglicine) (0.05–4 µM), and HT (hypotaurine) (1–32 µM) were applied as H_2_S donor, H_2_S biosynthesis inhibitor, and H_2_S scavenger, respectively [18]. All the reagents at analytical purity were obtained from China National Pharmaceutical Group Co., Ltd. (Sinopharm Chemical Reagent Co., Ltd., Beijing, China).

We designed several experiments in this study. First, the roots were exposed to Hg, or Hg + Se, respectively, for 72 h, followed by measuring root length and endogenous H_2_S level. Second, the root length were monitored at 6, 12, 24, 48, and 72 h, respectively, upon Hg + Se exposure. Third, roots were pretreated with NaHS, PAG, or HT, respectively, for 6 h. Then roots were exposed to Hg + Se for another 72 h, followed by measuring root length. This helped us to identify the proper concentration of NaHS (2 mM), PAG (0.5 µM), and HT (4 µM) for the following experiment. Fourth, roots were pretreated with PAG + NaHS or HT + NaHS, respectively, for 6 h. Then roots were exposed to Hg + Se for another 72 h, followed by measuring root length, root tip endogenous H_2_S level, root tip endogenous ROS level, and root tip cell death.

### 3.2. Evaluation of Endogenous H_2_S Level in Root Tip

Fluorescent probe WSP-1 (3′-methoxy-3-oxo-3H-spiro[isobenzofuran-1, 9′-xanthen]-6′-yl 2-(pyridin-2-yldisulfanyl)benzoate) was applied to track root tip endogenous H_2_S in situ [45,46]. Roots were incubated in WSP-1 solution (15 µM) at 25 °C for 40 min followed by rinsing with distilled water. Then root tips were observed and captured with a fluorescent microscope (ECLIPSE, TE2000-S, Nikon, Melville, NY, USA) with setting of excitation 480–490 nm/emission 515–525 nm.

### 3.3. Evaluation of Total ROS Level in Root Tip

Fluorescent probe DCFH-DA (2′,7′-dichlorofluorescein diacetate) was applied to detect root tip ROS in situ [29]. Roots were incubated in DCFH-DA solution (10 µM) at 25 °C for 10 min followed by rinsing with distilled water. Then root tips were observed and captured with a fluorescent microscope (ECLIPSE, TE2000-S) with setting of excitation 480–490 nm/emission 515–525 nm.

### 3.4. Evaluation of Cell Death in Root Tip

Fluorescent probe PI was used to detect root tip cell death in situ [47]. Roots were incubated in PI (propidium iodide) solution (20 µM) at 25 °C for 20 min followed by rinsing with distilled water. Then the root tips were observed and captured with a fluorescent microscope (ECLIPSE, TE2000-S) with setting of excitation 530–540 nm/emission 610–620 nm.

### 3.5. Evaluation of Oxidative Injury in Root Tip

Schiff’s reagent was used to detect root lipid peroxidation in situ [48]. Roots were stained with Schiff’s reagent for 20 min followed by rinsing with 0.5% (*w/v*) K_2_S_2_O_5_ (prepared in 0.05 M of HCl) until the color became light red. Then roots were photographed with a digital camera.

Evans blue was used to detect loss of membrane integrity in root cells in situ [49]. Roots were stained with Schiff’s reagent (0.025%, *w/v*) for 20 min followed by rinsing with distilled water. Then roots were photographed with a digital camera.

### 3.6. Statistical Analysis

Each result was presented as mean ± SD (standard deviation) of at 3–10 replicates. Least significant difference test (LSD) was performed on data following ANOVA tests to test for significant (*p* < 0.05) differences among treatments. The significant difference at *p* < 0.05 level between two designated treatments was compared using ANOVA with *F* test.

## Figures and Tables

**Figure 1 ijms-23-02854-f001:**
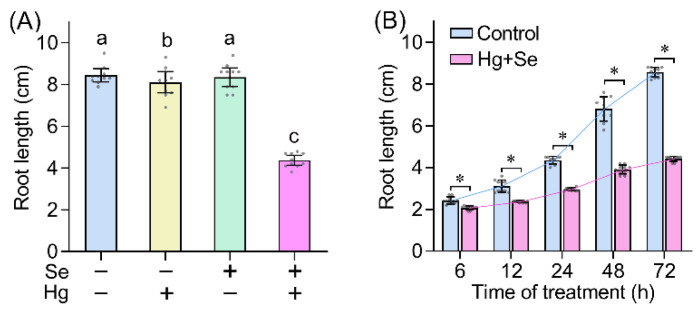
Hg + Se inhibited root growth of *B. rapa* seedlings. (**A**) The effect of Se, Hg, or Se + Hg on root length. Different lowercase letters indicated significant difference among different treatments (one-way analysis of variance, ANOVA; *n* = 10; *p* < 0.05). (**B**) Time-course monitoring of root length upon Se + Hg treatment. Asterisk indicated significant difference between control and Se + Hg (ANOVA; *n* = 10; *p* < 0.05).

**Figure 2 ijms-23-02854-f002:**
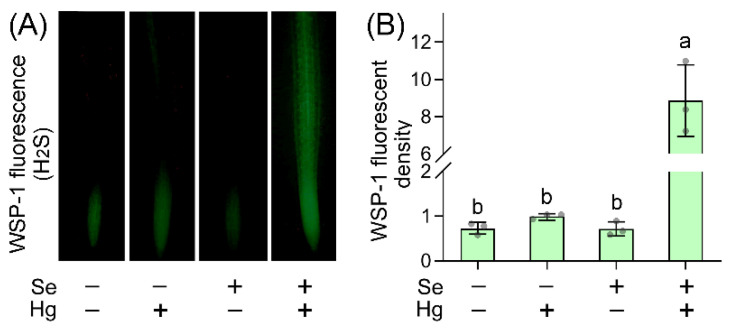
Hg + Se enhanced endogenous H_2_S level in the root tip of *B. rapa* seedlings. (**A**) Endogenous H_2_S in root tip was indicated by WSP-1 fluorescence. (**B**) Calculated relative WSP-1 fluorescent density in root tips. Different lowercase letters indicated significant difference among different treatments (ANOVA; *n* = 3; *p* < 0.05).

**Figure 3 ijms-23-02854-f003:**
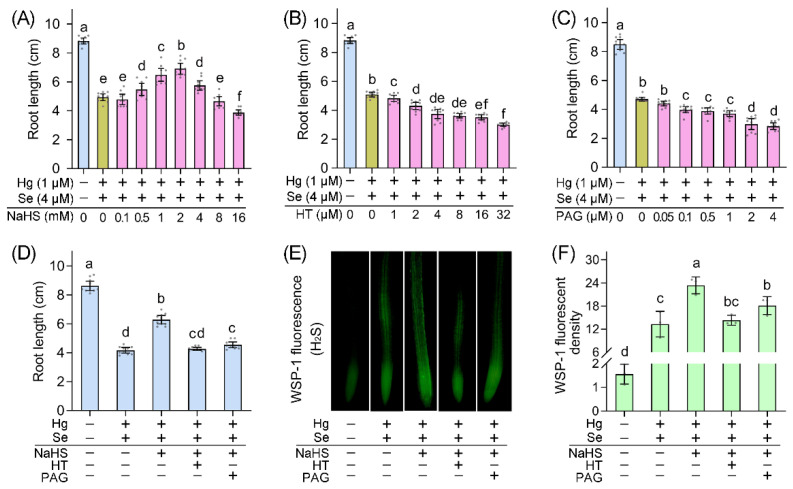
Endogenous H_2_S is involved in recovering root growth of *B. rapa* seedlings under Hg + Se stress. (**A**) Effect of pretreatment with NaHS on root growth under Hg + Se stress. (**B**) Effect of pretreatment with HT on root growth under Hg + Se stress. (**C**) Effect of pretreatment with PAG on root growth under Hg + Se stress. (**D**) Effect of combined pretreatment of NaHS, HT, and PAG on root growth under Hg + Se stress. € Effect of combined pretreatment of NaHS, HT, and PAG on endogenous H_2_S level indicated by WSP-1 fluorescence in root tips. (**F**) Calculated relative WSP-1 fluorescent density in root tips with respect € (**E**). Different lowercase letters in (**A**–**D**,**F**) indicated significant difference among different treatments (ANOVA; *n* = 10 or 3; *p* < 0.05).

**Figure 4 ijms-23-02854-f004:**
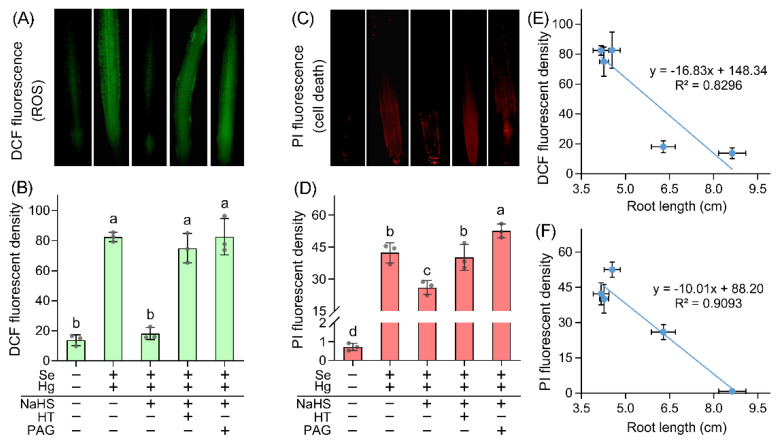
Endogenous H_2_S is involved in decreasing ROS accumulation and alleviating cell death in the root tip of *B. rapa* seedlings upon Hg + Se stress. (**A**) In situ detection of ROS accumulation with DCF fluorescence in root tip. (**B**) Calculated relative DCF fluorescent density in root tips. (**C**) In situ detection of ROS accumulation with PI fluorescence in root tip. (**D**) Calculated relative PI fluorescent density in root tips. (**E**) Correlation analysis between root length and DCF fluorescent density. (**F**) Correlation analysis between root length and PI fluorescent density. Different lowercase letters in (**B**,**D**) indicated significant difference among different treatments (ANOVA; *n* = 3; *p* < 0.05).

**Figure 5 ijms-23-02854-f005:**
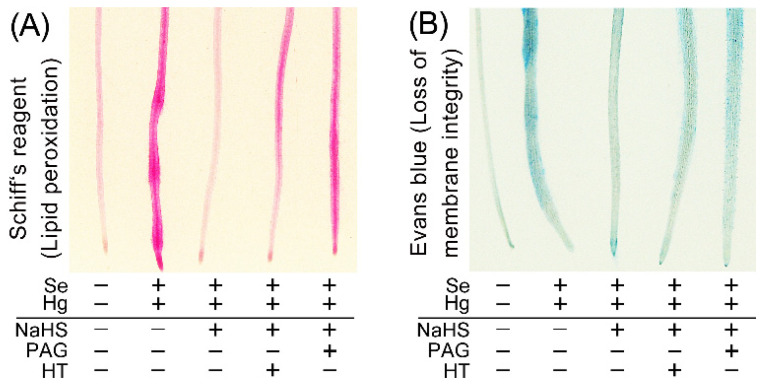
Endogenous H_2_S is involved in alleviating oxidative injury in the roots of *B. rapa* seedlings upon Hg + Se stress. (**A**) In vivo detection of lipid peroxidation in roots. (**B**) In vivo detection of loss of membrane integrity in roots.

## Data Availability

Not applicable.

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
