# Peer review of "The Defensive Role of Endogenous H2S in Brassica rapa against Mercury-Selenium Combined Stress"

_ijms, 2022, doi:10.3390/ijms23052854_

Round 1

Reviewer 1 Report

The work is devoted to the study of the contribution of hydrogen sulfide, a gas transmitter, to the realization of resistance in the roots of Brassica rapa L.  plants under the combined action of mercury -selenium stress. The abstract is quite clear. In the Introduction part, link [9] is not really about plants, please change it. In terms of results and discussion, Fig. 1A is not entirely clear for perception, it needs to be made more understandable.

In addition, please focus on what is the mechanism of the negative effect of the combined use of mercury and selenium, in addition to explaining that they cause the accumulation of ROS [10], at the same time, additional accumulation of hydrogen sulfide under the action of NaHS contributed to overcoming stress.

Please specify in the Materials and Methods, these seedlings were etiolated, at what stage of ontogenesis were they used? It is well known that the ability to respond to stress is not only dose dependent but also varies with plant age.

The article needs a little improvement, while I want to note the high methodological level of this work.

Reviewer 2 Report

Authors provided new evidence about the role of endogenous H2S in response to combined stress of mercury and selenium. The topic is up-to-date and could be interesting for many researcher and stakeholders in wide field.

Introduction is well structured, however there is a lack of toxic level and average contents because these values help us to understand the importance of this study and explain the used values in methods.

Beginning with "And" is not the best in English language, so please avoid to use this word.

In Line 43, EC number of enzyme is missing.

In the Results and Discussion, authors mentioned the results of single treatments, however only the combined treatment can be seen in Fig 1B, so please rewrite those sentence emphasizing that readers only could see the results of combined treatments, or write data not shown. Please check the English tense in case of L133 and L140.

My question is that authors checked only the root length or they could investigate any other parameters of roots, eg. root hairs or lateral roots?

In Materials and Methods, from L198 to L206 is not so clear, so please rewrite and describe the treatments to be easily followed by readers. Also, please write about the selection of the precise concentrations. I think it would be better to write again in Methods the full name of compounds or probes like WSP-1. 

Overall, this manuscript is well written and making corrections could be help to be accepted. I recommend this manuscript to be accepted after minor revisions.

Author Response

Responses to Reviewer #2

  1. Introduction is well structured, however there is a lack of toxic level and average contents because these values help us to understand the importance of this study and explain the used values in methods.

Response: Thanks for your suggestion. We have supplied the average content and toxic level of Hg as well as the range of Se concentration with beneficial effect in the part of “Introduction” in revised manuscript.

  1. Beginning with "And" is not the best in English language, so please avoid to use this word.

Response: Thanks for your suggestion. We have double-checked the whole manuscript to revise it.

  1. In Line 43, EC number of enzyme is missing.

Response: Thanks for your suggestion. We have supplied the EC number for DES in revised manuscript.

  1. In the Results and Discussion, authors mentioned the results of single treatments, however only the combined treatment can be seen in Fig 1B, so please rewrite those sentence emphasizing that readers only could see the results of combined treatments, or write data not shown. Please check the English tense in case of L133 and L140.

Response: Thanks for your suggestion. We have revised the description for Fig.1B in order to guide the readers to focus on the combined treatment in Fig. 1B. In addition, the English tense has been checked and revised in revised manuscript. 

  1. My question is that authors checked only the root length or they could investigate any other parameters of roots, eg. root hairs or lateral roots?

Response: Thanks for your suggestion. We used 3-days old seedlings to start the experiment. Then the seedlings were treated for another 3 days (72 h). The plant age is pretty young, which is sensitive to stress conditions. The seedling in this growth period only has primary root without lateral root yet. So it is easier to evaluate root growth by only measuring the length of primary root. We have supplied the detailed information about it in the first part of Materials and Methods in revised manuscript. We tend to use bigger plants with prolonged treatment in order to evaluate the changes of root morphogenesis (including primary root, lateral root, and root hair) in further studies.

  1. In Materials and Methods, from L198 to L206 is not so clear, so please rewrite and describe the treatments to be easily followed by readers. Also, please write about the selection of the precise concentrations. I think it would be better to write again in Methods the full name of compounds or probes like WSP-1.

Response: Thanks for your suggestion. We have revised the description of this part in order to make it easier to follow the experimental setup and the selection of treatment concentrations. And the full name of compounds used in this study have been written again in Methods in revised manuscript.